

# The feasibility of predicting ground reaction forces during running from a trunk accelerometry driven mass-spring-damper model

Niels J. Nedergaard[1,2], Jasper Verheul[1], Barry Drust[1], Terence Etchells[3], Paulo Lisboa[3], Mark A. Robinson[1] and Jos Vanrenterghem[1,2]

[1] Research Institute for Sport and Exercise Sciences, Liverpool John Moores University, Liverpool, United Kingdom
[2] Department of Rehabilitation Sciences, Katholieke Universiteit Leuven, Leuven, Belgium
[3] Department of Applied Mathematics, Liverpool John Moores University, Liverpool, United Kingdom

Corresponding author
Jos Vanrenterghem,
jos.vanrenterghem@kuleuven.be

## ABSTRACT

**Background.** Monitoring the external ground reaction forces (GRF) acting on the human body during running could help to understand how external loads influence tissue adaptation over time. Although mass-spring-damper (MSD) models have the potential to simulate the complex multi-segmental mechanics of the human body and predict GRF, these models currently require input from measured GRF limiting their application in field settings. Based on the hypothesis that the acceleration of the MSD-model's upper mass primarily represents the acceleration of the trunk segment, this paper explored the feasibility of using measured trunk accelerometry to estimate the MSD-model parameters required to predict resultant GRF during running.

**Methods.** Twenty male athletes ran at approach speeds between 2–5 m s$^{-1}$. Resultant trunk accelerometry was used as a surrogate of the MSD-model upper mass acceleration to estimate the MSD-model parameters ($ACC_{param}$) required to predict resultant GRF. A purpose-built gradient descent optimisation routine was used where the MSD-model's upper mass acceleration was fitted to the measured trunk accelerometer signal. Root mean squared errors (RMSE) were calculated to evaluate the accuracy of the trunk accelerometry fitting and GRF predictions. In addition, MSD-model parameters were estimated from fitting measured resultant GRF ($GRF_{param}$), to explore the difference between $ACC_{param}$ and $GRF_{param}$.

**Results.** Despite a good match between the measured trunk accelerometry and the MSD-model's upper mass acceleration (median RMSE between 0.16 and 0.22 g), poor GRF predictions (median RMSE between 6.68 and 12.77 N kg$^{-1}$) were observed. In contrast, the MSD-model was able to replicate the measured GRF with high accuracy (median RMSE between 0.45 and 0.59 N kg$^{-1}$) across running speeds from $GRF_{param}$. The $ACC_{param}$ from measured trunk accelerometry under- or overestimated the $GRF_{param}$ obtained from measured GRF, and generally demonstrated larger within parameter variations.

**Discussion.** Despite the potential of obtaining a close fit between the MSD-model's upper mass acceleration and the measured trunk accelerometry, the $ACC_{param}$ estimated from this process were inadequate to predict resultant GRF waveforms during slow to

moderate speed running. We therefore conclude that trunk-mounted accelerometry alone is inappropriate as input for the MSD-model to predict meaningful GRF waveforms. Further investigations are needed to continue to explore the feasibility of using body-worn micro sensor technology to drive simple human body models that would allow practitioners and researchers to estimate and monitor GRF waveforms in field settings.

## INTRODUCTION

Humans generate considerable forces against the ground during running to maintain an upright posture. This comes at the cost of equal and opposite ground reaction forces (GRF) acting on the body during every foot-ground contact (*Cavanagh & Lafortune, 1980*). These GRF put the body's soft tissues (e.g., bones, cartilage, muscles, tendons and ligaments) under biomechanical stresses which over time are expected to lead to beneficial structural adaptations (*Kibler, Chandler & Stracener, 1992*; *Dye, 2005*). Inadequate recovery or repetitive GRF with excessive magnitudes can instead lead to negative adaptions and tissue damage (*Kibler, Chandler & Stracener, 1992*; *Dye, 2005*). The ability to monitor an athlete's GRF during running can therefore help to better understand the relationship between the external forces experienced and soft-tissue adaptations (*Vanrenterghem et al., 2017*) ultimately helping to prevent musculoskeletal injury.

Accurate monitoring of GRF waveforms during running is currently restricted to laboratory environments where GRF waveforms are measured with force platforms built into the ground, or derived from whole-body kinematics (*Bobbert, Schamhardt & Nigg, 1991*; *Winter, 2005*). With recent developments of low-cost sensor based micro technology (*Camomilla et al., 2018*), accelerometry has become a popular tool to evaluate running mechanics outside laboratory environments in long and middle distance running (*Tao et al., 2012*) and professional team sports (*Akenhead & Nassis, 2016*). Accelerometry also offers opportunities to estimate loading related GRF characteristics (e.g., *Lafortune, 1991*; *Wundersitz et al., 2013*; *Neugerbauer, Collins & Hawkins, 2014*; *Raper et al., 2018*), and tibia-mounted accelerometry has for example been used as surrogate measure of peak GRF since the early 90s (*Lafortune, 1991*; *Lafortune, Lake & Hennig, 1995*). However, recent studies found weak to moderate linear relationships between peak accelerations measured from body-worn accelerometry (trunk- and tibia-mounted accelerometers) and peak whole-body accelerations measured from force platforms during running (*Wundersitz et al., 2013*; *Nedergaard et al., 2017a*; *Raper et al., 2018*). Since body-worn accelerometers only measure segmental acceleration, the use of a single accelerometer has to date been inadequate to incorporate the complex multi-segmental accelerations that result in task-specific GRF patterns (*Nedergaard et al., 2017a*). Recent studies have indicated that from the combination of three or more body-worn inertial sensors and machine learning one can

estimate GRF and knee joint moments with reasonable accuracy during running related locomotion (*Johnson et al., 2018*; *Wouda et al., 2018*), but the broader application of such approaches is constrained by the requirement of multiple sensors, machine learning tools, and large data sets. Therefore, if it were possible to estimate accurate GRF waveforms from a single body-worn sensor, it would provide practitioners and researchers with a useful tool to monitor the biomechanical load in field settings.

Since the overall motion of the human body has a spring-like behaviour during running, simple mass–spring models, consisting of a single mass and spring, have been widely used to estimate the vertical GRF in field settings (e.g., *Alexander, 1984*; *Blickhan, 1989*; *McMahon & Cheng, 1990*). Moreover, such models have been used in combination with trunk-mounted accelerometry to estimate the required model parameters (*Gaudino et al., 2013*; *Buchheit, Gray & Morin, 2015*). Unfortunately, the initial high-frequency impact peak typically observed in the GRF waveform during running, which is speculated to be linked with negative tissue adaptations and risk of injury (*Nigg, Cole & Bruggemann, 1995*; *Hreljac, Marshall & Hume, 2000*; *Milner et al., 2006*), cannot be estimated with this oversimplified model (*Alexander, Bennett & Ker, 1986*; *Bullimore & Burn, 2007*). A more complex mass–spring-damper model (MSD-model) better replicates the GRF waveforms for running at moderate speeds (3.83 m s$^{-1}$ ± 5%), including both impact and active peaks (*Derrick, Caldwell & Hamill, 2000*). This model consists of a lower mass connected to a spring in parallel with a damper, representing the support leg during foot-ground contact, and an upper mass and spring representing the dynamics of the rest of the body. However, the ability to use trunk-mounted accelerometry to estimate the required model parameters for this model is yet unknown.

The aim of this study was to examine if the acceleration of the MSD-model's upper mass represents the acceleration of the trunk segment measured with trunk-mounted accelerometry during running. This hypothesis seems feasible, since the trunk segment represents half of the body mass (*Dempster, 1955*). If this provides a reasonable approximation, it might be feasible to estimate the required MSD-model parameters from trunk accelerometry to subsequently predict GRF from the MSD-model behaviour. Specifically, we therefore explored (1) the feasibility to estimate the MSD-model's eight natural model parameters from measured trunk accelerometry, and (2) whether these model parameters in fact predict reasonably accurate GRF waveforms during running at slow to moderate running speeds.

## MATERIALS & METHODS
### Subjects and protocol
Twenty healthy male athletes (age 22 ± 4 years, height 178 ± 8 cm, mass 76 ± 11 kg) who engaged in running related sports activities on a weekly basis volunteered to participate in this study. The institutional ethics committee at Liverpool John Moores University granted ethical approval for this study (ethics approval number: 09/SPS/010) in accordance with the Declaration of Helsinki, and written consent was obtained from all participants. After a 15 min warm-up (including light jogging, dynamic stretching and individual dynamic

tasks) and an individual number of familiarisation trials, the participants were asked to run over a force platform at different running speeds of 2, 3, 4 and 5 m s$^{-1}$ ($\pm5\%$) in a randomised condition order. Running speeds were measured with photocell timing gates (Brower Timing System, Utah, USA) placed 2 m apart, with the last gate positioned 2 m before the centre of the force platform as described in *Vanrenterghem et al. (2012)*. The participants completed four trials of each running speed, landing on the force platform with their dominant leg (defined as the self-reported preferred kicking leg (*Van Melick et al., 2017*)). Trials with unsuccessful foot contacts on the force platform (double foot contact or when the foot was not placed within the force platform) and/or when the desired approach speed was not achieved were repeated until a valid trial was recorded.

**Experimental measurements**
Resultant ground reaction forces were measured (*GRF*) with a sampling frequency of 3,000 Hz from a 0.9 $\times$ 0.6 m$^2$ Kistler force platform (9287C, Kistler Instruments Ltd., Winterthur, Switzerland). Resultant trunk accelerations (TrunkAcc) were simultaneously collected at 100 Hz using a tri-axial accelerometer (KXP94, Kionex, Inc., Ithaca, NY, USA) with an output range of $\pm13$ g embedded within a commercial GPS device (MinimaxX S4, Catapult Innovations, Scoresby, Australia) with a total weight of 67 grams and 88 $\times$ 50 $\times$ 19 mm in dimension. The GPS device was positioned on the dorsal part of the upper trunk between the scapulae within a small pocket of a tight fitted elastic vest according to the manufacturer's recommendations (*Boyd, Ball & Aughey, 2011*). Different vest sizes were used to ensure the tightest fit for the individual participants. TrunkAcc data (measured in the units g) was pre-processed with the manufacturer's proprietary filter algorithms (50 Hz low-pass filter, personal communication with the manufacturer), and downloaded as 'raw accelerometer data' from the manufacturer's software (Catapult Sprint, version 5.1.7, Melbourne, Australia) after each test session. Each session also included a static measurement at the beginning and end of the session to detect any calibration drift over time, and none was detected. TrunkAcc and GRF were synchronised using a combination of time and event synchronisation as described in *Nedergaard et al. (2017a)* and exported to Matlab (version R2016a, The MathWorks, Inc., Natick, MA, USA) where a 4th order recursive Butterworth low-pass filter with a cut-off frequency of 20 Hz was applied to GRF and TrunkAcc. GRF data was collected from a single stance phase per trial, where touch down and take off were defined when the vertical GRF crossed a 20 N threshold.

**Mass-spring-damper model**
The complex multi-segmental dynamics of the human body during stance phase were modelled as a passive MSD-model (*Alexander, Bennett & Ker, 1986*; *Derrick, Caldwell & Hamill, 2000*). This model consists of two masses (Fig. 1); a lower point mass ($m_2$) on top of a linear spring ($k_2$) in parallel with a damper ($c$) representing the support leg; an upper point mass ($m_1$) representing the dynamics of the rest of the body and another linear spring ($k_1$) connecting the two masses.

The one-dimensional motion of the MSD-model was described by the acceleration of the two masses Eqs. (5) and (6):

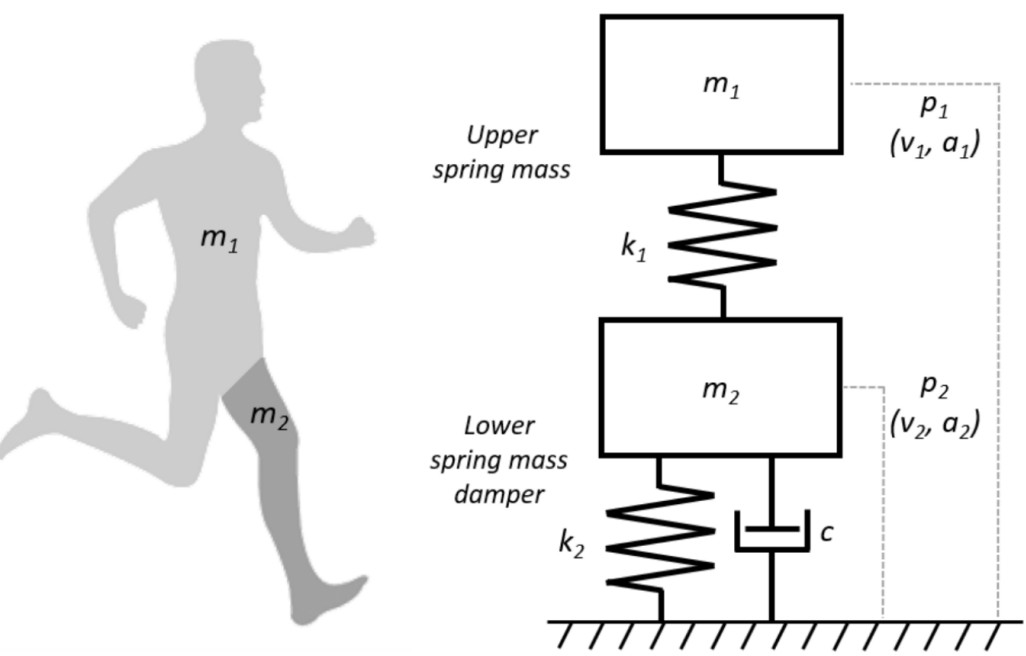

**Figure 1 An illustration of the human body represented as a MSD-model.** The MSD-model consists of a lower mass spring damper element ($m_2$, $k_2$, c) representing the support leg of the human body and an upper mass spring element ($m_1$, $k_1$) representing the rest of the human body.

$$\lambda = \frac{m_1}{m_2} \tag{1}$$

$$\omega_1^2 = \frac{k_1}{m_1} = \frac{(1+\lambda)k_1}{\lambda M} \tag{2}$$

$$\omega_2^2 = \frac{k_2}{m_2} = \frac{(1+\lambda)k_2}{M} \tag{3}$$

$$\zeta = \frac{c}{2\sqrt{k_2 m_2}} \tag{4}$$

$$a_1 = -\omega_1^2 (p_1 - p_2) + g \tag{5}$$

$$a_2 = -\omega_2^2 p_2 + \omega_1^2 \lambda (p_1 - p_2) - 2\zeta \omega_2^2 v_2 + g \tag{6}$$

$$GRF_{model} = \frac{M\omega_2}{1+\lambda}(\omega_2 p_2 + 2\zeta v_2) \tag{7}$$

where $p_1$, $p_2$ $v_1$, $v_2$, $a_1$, and $a_2$ are the initial positions, velocities and, accelerations of the two masses ($m_1$ and $m_2$), respectively; $\lambda$ is the mass ratio of the lower mass relative to the total body mass (Eq. 1); $\omega_1^2$ and $\omega_2^2$ are the natural frequencies of the springs (Eqs. (2) and (3)) based on the linear spring constants ($k_1$ and $k_2$) and the mass of the two masses ($m_1$ and $m_2$); $\zeta$ is the damping ratio of the damper (Eq. 4); and $g$ is the acceleration from gravitational acceleration ($-9.81$ m s$^{-1}$). The resultant GRF acting on the MSD-model is calculated as in Eq. (7), where M is the sum of the two masses (i.e., total body mass):

## Model parameter estimation

To estimate the eight MSD-model parameters ($p_1$, $p_2$ $v_1$, $v_2$, $\omega_1^2$ $\omega_2^2$, $\zeta$, $\lambda$), we used gravity corrected TrunkAcc from the stance phase as a surrogate of the MSD-model's upper mass acceleration (Fig. 2A). For each trial, model parameters ($ACC_{param}$) were optimised by fitting the MSD-model's upper mass acceleration ($a_1$) to the TrunkAcc signal. A purpose-built gradient descent optimisation routine in Matlab was used, where the two second-order differential equations of the MSD-model's motion Eqs. (5) and (6) were transformed to four first-order differential equations and solved numerically with a Runge Kutta 4th order method. Root mean squared error (RMSE) between the TrunkAcc and $a_1$ waveforms were calculated for every iteration to determine the optimal model $ACC_{param}$ combination that best fitted TrunkAcc for the individual trials. The $ACC_{param}$ estimated from the TrunkAcc fitting were then used to predict the resultant GRF from Eq. (7). Furthermore, to help understand differences in estimated model parameters and the predicted versus measured resultant GRF, we also estimated the eight model parameters ($GRF_{param}$) by fitting the MSD-model to the measured GRF (Fig. 2B), similar to the approach previously described by *Derrick, Caldwell & Hamill (2000)*.

## Statistical analysis

Measured and modelled GRF were normalised to the participants' mass. RMSE between the TrunkAcc and $a_1$, waveforms, and between the measured GRF and predicted GRF waveforms, were calculated to evaluate the accuracy of the TrunkAcc fitting and the predicted GRF, respectively. RMSE median and interquartile range (25th to 75th percentile) were calculated to determine the variation in the model's accuracy within and across running speeds. Similarly, the median and interquartile range (25th to 75th percentile) of the $ACC_{param}$ and $GRF_{param}$ were calculated to explore the variation within and across running speeds. The median data presented and discussed in the following is the median of all trials within the individual running speeds ($N = 80$ trials) and the overall median across all running speeds ($N = 320$ trials).

## RESULTS

The first step was to estimate the required $ACC_{param}$ that fit the MSD-model's upper mass acceleration to the measured TrunkAcc signal. The MSD-model was able to fit the measured TrunkAcc with good accuracy across running speeds, though $a_1$ systematically underestimated the magnitude of the first peak observed in the accelerometry signal (Fig. 3A). The median RMSE (interquartile range 25th to 75th percentile) of the TrunkAcc fitting increased from 0.16 (0.12; 0.22) g at the slowest running speed to between 0.21 (0.16; 0.26) g and 0.22 (0.16; 0.30) g for three faster running speeds. Though similar median RMSE values were observed across the three fastest running speeds, the interquartile range increased with increased running speeds (Fig. 3C). Despite the good match between $a_1$ and TrunkAcc, poor GRF predictions were observed across running speeds (Fig. 3B) and the median RMSE of the predicted GRF systematically increased with running speeds, from 6.68 (3.81; 15.30) N $kg^{-1}$ at 2 m $s^{-1}$ to 12.77 (7.78; 27.22) N $kg^{-1}$ at 5 m $s^{-1}$.

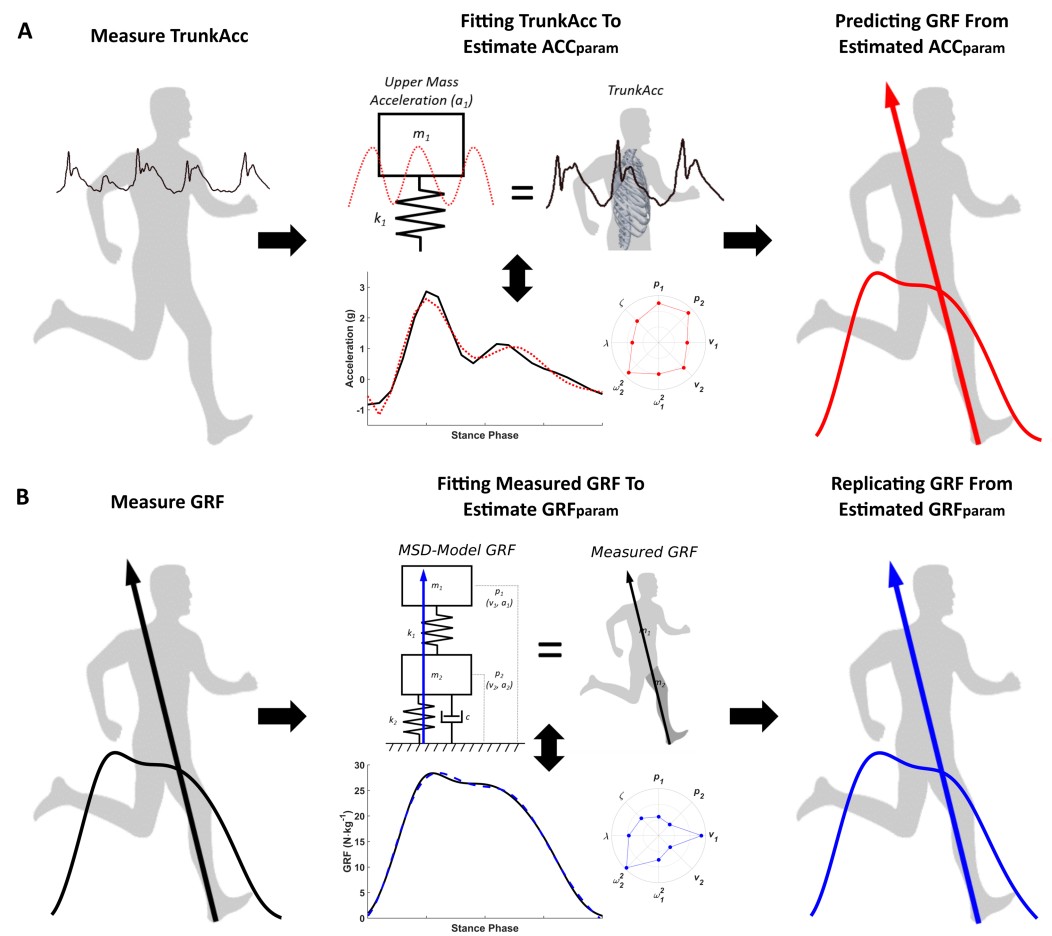

**Figure 2  Estimating MSD-model parameters by fitting the MSD-model to measured trunk accelerometry and measured GRF.** (A) illustrates the trunk driven MSD-model where measured trunk accelerometry (TrunkAcc) for the stance phase, is used to estimate the eight $ACC_{param}$, based on the hypothesis that the MSD-model's upper mass acceleration ($a_1$) primarily represents TrunkAcc, before GRF is calculated from the $ACC_{param}$ that best fitted TrunkAcc. (B) displays the traditional MSD- model approach, where the eight $GRF_{param}$ are estimated by fitting the model's GRF to the measured GRF.

Since the $ACC_{param}$ resulted in poor GRF predictions, we next estimated the $GRF_{param}$ by fitting the MSD-model to the measured GRF waveforms (Fig. 2B) to determine if there was any difference between the two sets of model parameters (Table 1) and to compare the upper mass acceleration to the measured TrunkAcc. The MSD-model was able to replicate the measured GRF with high accuracy when $GRF_{param}$ were estimated to directly fit the measured GRF (Fig. 4B). This was reflected in the low RMSE median and interquartile ranges observed across all running speeds (2 m s$^{-1}$: 0.45 (0.36; 0.60); 3 m s$^{-1}$: 0.47 (0.37; 0.61); 4 m s$^{-1}$: 0.53 (0.39; 0.66); 5 m s$^{-1}$: 0.59 (0.46; 0.73); All Speeds: 0.51 (0.39; 0.64) N kg$^{-1}$). However, the MSD-model's upper mass acceleration profiles then deviated considerably from the acceleration profiles measured with trunk accelerometry (Fig. 4A). The $GRF_{param}$ also differed considerably from the $ACC_{param}$ (Figs. 4C and 4D). Namely, the $GRF_{param}$ demonstrated smaller within parameter variation, which was especially
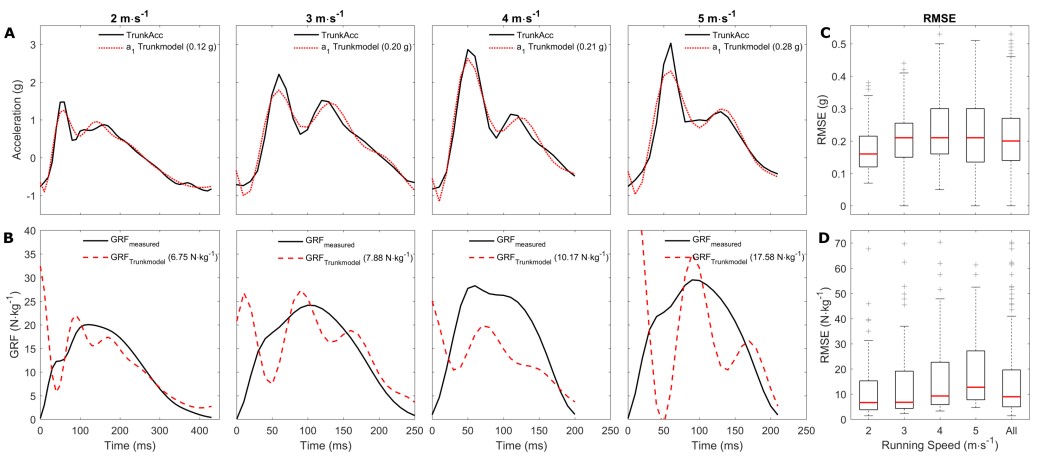

**Figure 3** **Representative examples of the trunk accelerometry fitting and GRF prediction, and the median RMSE across running speeds.** Representative examples of a single stride from multiple subjects. (A) display the fitting of the upper mass acceleration to the trunk accelerometry signal across running speeds, and (B) display the measured and predicted GRF for the same trials. The RMSE for the trunk accelerometry fitting and GRF predictions are displayed in brackets for the individual examples. (C and D) display the average RMSE median, and 25th and 75th interquartile range for the trunk accelerometry fitting and GRF prediction respectively within and across the individual running speeds. A total of 17 extreme outliers (3 m s$^{-1}$: 3; 4 m s$^{-1}$ : 7; 5 m s$^{-1}$ : 7 outliers) were removed through visual inspection from the boxplots in (D) to improve the visual interpretation.

evident for $p_2$ and $v_2$. Also, lower $v_1$ (median difference 0.47 m s$^{-1}$) and higher $v_2$ (median difference $-1.73$ m s$^{-1}$) values were observed across running speeds.

## DISCUSSION

This study illustrates that the MSD-model's upper mass acceleration could be fitted to the measured trunk accelerometry with high accuracy, but the ACC$_{param}$ estimated from this process did not lead to accurate predictions of resultant GRF waveforms across a range of slow to moderate running speeds. Further analysis of the MSD-model behaviour when fitting to the measured resultant GRF revealed a considerable discrepancy in GRF$_{param}$ compared to the ACC$_{param}$ when fitting the MSD-model to measured trunk accelerometry signals. These results demonstrate that our initial hypothesis that the MSD-model's upper mass acceleration primarily represents the acceleration of the trunk was false.

### Model parameter estimation

The eight model parameters are fundamental to calculating the resultant GRF acting on the MSD-model, and though fitting TrunkAcc was successful, the ACC$_{param}$ estimated from this approach resulted in poor GRF predictions. Based on the equation of the upper mass acceleration (Eq. 5) and the ACC$_{param}$ estimated from TrunkAcc, it seems that the MSD-model was able to fit the TrunkAcc by keeping the initial position of the upper mass ($p_1$) and lower mass ($p_2$) low, and by keeping the spring stiffness of the upper spring ($\omega_1{}^2$) low. Whereas $p_1$ has minor influence on the predicted GRF, the velocity of the upper mass at initial contact ($v_1$) is indirectly influenced by changes in the initial upper mass position

**Table 1 Average median, 25th and 75th interquartile range for the $ACC_{param}$ and $GRF_{param}$ within and across the individual running speeds.**

| Model Parameters | 2 (m s$^{-1}$) | 3 (m s$^{-1}$) | 4 (m s$^{-1}$) | 5 (m s$^{-1}$) | All |
|---|---|---|---|---|---|
| | Median (25th; 75th) | Median (25th; 75th) | Median (25th; 75th ) | Median (25th; 75th) | Median (25th; 75th) |
| **$p_1$ (m)** | | | | | |
| $ACC_{param}$ | −0.02 (−0.04; −0.01) | −0.01 (−0.04; 0.00) | −0.02 (−0.05; 0.00) | −0.03 (−0.06; −0.01) | −0.02 (−0.05; −0.01) |
| $GRF_{param}$ | 0.00 (−0.01; 0.00) | 0.00 (−0.01; 0.00) | 0.00 (−0.02; 0.00) | −0.01 (−0.02; −0.01) | −0.01 (−0.02; 0.00) |
| **$p_2$ (m)** | | | | | |
| $ACC_{param}$ | −0.01 (−0.02; 0.00) | 0.00 (−0.02; 0.01) | −0.01 (−0.03; 0.00) | −0.01 (−0.04; 0.00) | −0.01 (−0.03; 0.00) |
| $GRF_{param}$ | 0.00 (0.00; 0.01) | 0.00 (0.00; 0.01) | 0.00 (0.00; 0.00) | 0.00 (0.00; 0.00) | 0.00 (0.00; 0.00) |
| **$v_1$ (m s$^{-1}$)** | | | | | |
| $ACC_{param}$ | −0.58 (−0.65; −0.50) | −0.67 (−0.82; −0.60) | −0.71 (−0.82; −0.60) | −0.60 (−0.69; −0.41) | −0.64 (−0.75; −0.54) |
| $GRF_{param}$ | −0.91 (−1.12; −0.72) | −1.04 (−1.26; −0.92) | −1.24 (−1.34; −1.11) | −1.13 (−1.27; −0.98) | −1.11 (−1.28; −0.92) |
| **$v_2$ (m s$^{-1}$)** | | | | | |
| $ACC_{param}$ | −1.98 (−2.71; −1.37) | −1.59 (−2.91; −0.93) | −1.78 (−3.13; −1.20) | −1.55 (−2.39; −0.60) | −1.75 (−2.76; −1.08) |
| $GRF_{param}$ | −0.02 (−0.40; 0.00) | −0.05 (−0.28; 0.00) | −0.01 (−0.24; 0.00) | 0.00 (−0.10; 0.00) | −0.02 (−0.25; 0.00) |
| **$\omega_1^2$ (N m$^{-1}$ kg$^{-1}$)** | | | | | |
| $ACC_{param}$ | 334 (233; 622) | 508 (233; 966) | 477 (315; 1,193) | 512 (331; 977) | 469 (267; 958) |
| $GRF_{param}$ | 528 (370; 721) | 577 (357; 935) | 621 (385; 959) | 687 (495; 1,025) | 604 (411; 899) |
| **$\omega_2^2$ (N m$^{-1}$ kg$^{-1}$)** | | | | | |
| $ACC_{param}$ | 2,537 (1,167; 3,895) | 2,584 (1,152; 4,174) | 2,967 (1,628; 4,688) | 3,460 (1,517; 5,094) | 2,795 (1,320; 4,362) |
| $GRF_{param}$ | 2,421 (1,516; 3,420) | 2,966 (2,265; 4,593) | 3574 (2,305; 5,094) | 4,006 (3,111; 6,550) | 3,253 (2,207; 4,894) |
| **$\lambda$ (au)** | | | | | |
| $ACC_{param}$ | 2.86 (1.72; 4.60) | 2.61 (1.46; 4.30) | 3.11 (1.06; 4.27) | 2.25 (0.97; 3.28) | 2.62 (1.28; 4.13) |
| $GRF_{param}$ | 4.02 (2.14; 6.62) | 5.19 (2.82; 6.51) | 2.84 (1.86; 5.90) | 2.74 (1.82; 3.40) | 3.32 (2.04; 5.84) |
| **$\zeta$ (au)** | | | | | |
| $ACC_{param}$ | 0.23 (0.18; 0.33) | 0.21 (0.15; 0.32) | 0.20 (0.15; 0.30) | 0.16 (0.07; 0.32) | 0.20 (0.15; 0.32) |
| $GRF_{param}$ | 0.38 (0.29; 0.58) | 0.39 (0.28; 0.51) | 0.37 (0.28; 0.45) | 0.31 (0.25; 0.40) | 0.36 (0.27; 0.45) |

($v_1 = \dot{p}_1$). *Derrick, Caldwell & Hamill (2000)* found that decreased $v_1$ has a large impact on the duration of the stance phase and therefore could have contributed to the overestimation of foot-ground contact (Fig. 3B). Similarly, the MSD-model decreased the spring stiffness of the upper spring ($\omega_1^2$) to better fit the two acceleration peaks typically observed in the TrunkAcc data, which has previously been shown to increase the duration of the stance phase (*Derrick, Caldwell & Hamill, 2000*). Furthermore, the MSD-model lowered the initial position of the lower mass ($p_2$), which previously has been shown to both increase the GRF at touch down and decrease the magnitude of the impact peak (*Derrick, Caldwell & Hamill, 2000*). We therefore believe that the high GRF values observed in our GRF predictions at touch down (Fig. 3B) were primarily related to the lower initial position of the lower mass ($p_2$) required to fit the upper mass acceleration to the TrunkAcc. Finally, the MSD-model also kept the damping ratio ($\zeta$) low to better fit the magnitude of the two acceleration peaks in the TrunkAcc. Decreasing the damping ratio, has however previously been shown to increase the oscillation in the model's GRF (*Alexander, Bennett & Ker, 1986*; *Derrick,*

Peer J

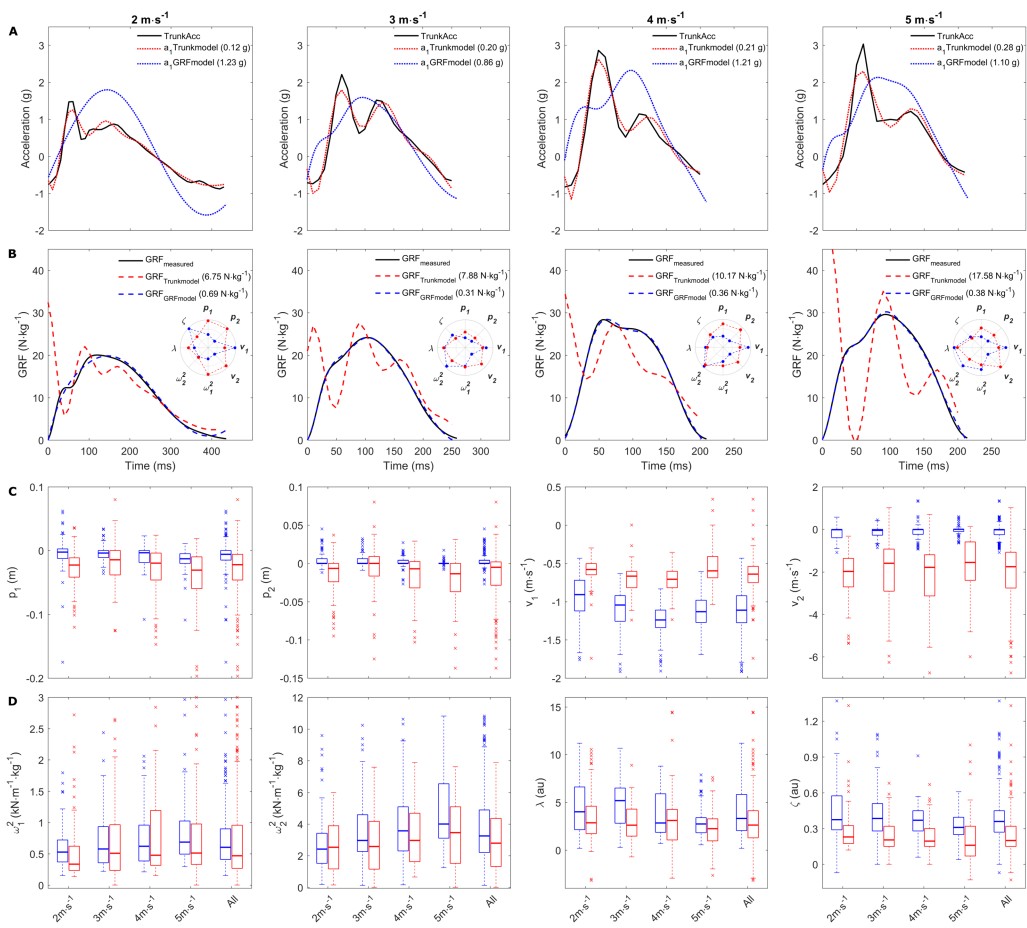

**Figure 4** **Representative examples of the upper mass acceleration, GRF and median ACC_param and GRF_param.** Representative examples of a single stride from multiple subjects. (A) display the measured trunk accelerometry and the MSD-model's upper mass acceleration, and (B) display the measured, predicted and replicated GRF. The RMSE for the trunk accelerometry fitting and GRF predictions are displayed in the brackets for the individual examples. The inserted polar plots display the estimated model parameters (in unscaled values) from the two approaches for the representative examples. (C and D) display the average median, 25th and 75th interquartile range for the ACC_param and GRF_param within and across the individual running speeds. A total of 33 extreme outliers were removed from the ACC_param ($p_1$: 7; $p_2$: 8; $v_1$: 2; $v_2$: 13; $\omega_1^2$: 1; $\lambda$: two outliers) and 15 extreme outliers were removed from the GRF_param ($v_1$: 6; $v_2$: 1; $\omega_2^2$: 3; $\lambda$: 5, $\zeta$: nine outliers) through visual inspection from the boxplots in (C and D) to improve the visual interpretation.

*Caldwell & Hamill, 2000*), and may therefore explain why our GRF predictions to a large extent include oscillating characteristics (Fig. 3B).

The comparison between the ACC_param estimated from the TrunkAcc and the GRF_param estimated from measured GRF, clearly demonstrates that the model is unsuitable for predicting GRF from TrunkAcc. A closer look at the GRF_param, showed that the median position and velocity of the lower mass ($p_2$ and $v_2$) was constant across running speeds and only varied marginally within running speeds (Fig. 4C). In addition, only small differences in median damping ratios ($\zeta$) were observed between running speeds in this study ($\zeta$

between 0.31 and 0.39 au). It was in fact kept constant ($\zeta = 0.35$ au) in the study by *Derrick, Caldwell & Hamill (2000)*. Based on these observations we explored the effect of keeping $p_2$, $v_2$, and $\zeta$ ACC$_{param}$ constant for all trials (using the median GRF$_{param}$ across running speeds), and for the remaining five MSD-model parameters use the trial specific ACC$_{param}$ to re-calculate the predicted GRF (Fig. S1). Whilst this decreased the variability of the GRF prediction (RMSE interquartile range) both within and across running speeds, only minor improvements were observed in the GRF prediction. This indicated that keeping selected ACC$_{param}$ constant would not substantially improve the GRF prediction in future studies. Furthermore, when selected ACC$_{param}$ were kept constant, their original interaction was broken.

## MSD-model hypothesis

If the trunk accelerometry data accurately represents the model's upper mass acceleration one would at least expect that the ACC$_{param}$ related to the motion and stiffness of the upper mass and spring ($p_1$, $v_1$, $\omega_1{}^2$) would be close to the GRF$_{param}$ estimated when fitting measured GRF. This was however not the case, and therefore naturally raises the questions as to whether the upper mass acceleration is equivalent to the acceleration measured from trunk accelerometry during running. The trunk accelerometry driven MSD-model approach introduced in this study is based on the hypothesis that the model's upper mass primarily represents the mass and motion of the trunk segment (*Alexander, Bennett & Ker, 1986*; *Derrick, Caldwell & Hamill, 2000*). Our results suggest however that this is not the case, and that independent accelerations of other body segments (e.g., the swing leg and arms) significantly contribute to the MSD-model's upper mass accelerations. We therefore conclude that the primary model hypothesis for this study was false, and that trunk-mounted accelerometry alone is inappropriate as input for the MSD-model to predict meaningful GRF waveforms.

A high initial peak related to the attenuation of the shock impact from the foot's collision with the ground (*Hamill, Derrick & Holt, 1995*; *Derrick, 2004*) dominated the TrunkAcc signals across running speeds. In contrast, a higher second peak related to the COM displacement during the stance phase dominated the upper mass acceleration when the MSD-model was fitted to measured GRF. This raised the technical question as to whether the poor GRF predictions observed from the measured accelerometer signal were partly a consequence of an artificially high frequency of that initial peak and whether the application of lower filter cut-off frequencies (cut-off frequencies of 20 Hz in the present study) would improve GRF predictions. To explore this, trunk accelerometry data of 10 representative participants was low-pass filtered with cut-off frequencies of 15, 10 and 5 Hz (Fig. S2). Whilst low cut-off frequencies (especially 10 and 5 Hz) to a large extent successfully removed the initial high-frequency peak in the accelerometry signal, and the RMSE between TrunkAcc and upper mass acceleration decreased, it only had a minor influence on the RMSE of the predicted GRF across running speeds (Fig. S2). Therefore, accelerometry post-processing did not improve the GRF predictions from TrunkAcc. This suggests that the trunk accelerometry signal in itself was not the main reason for the poor

GRF predictions, but rather an incorrect hypothesis that the MSD-model's upper mass acceleration primarily represents the acceleration of the trunk segment.

## Replicating GRF from measured GRF

Although TrunkAcc was unsuccessful in predicting GRF during running with a simple MSD-model, the MSD-model could successfully replicate measured GRF during slow to moderate running speeds. In fact, the inclusion of all eight GRF$_{param}$ in our optimisation routine, compared to only optimising the spring constants of the upper and lower springs ($k_1$ and $k_2$) and the position of the lower mass ($p_2$) (*Derrick, Caldwell & Hamill, 2000*) allowed us to replicate the measured GRF with higher accuracy. These findings illustrate that despite the MSD-model's simplicity it has the ability to replicate and potentially predict GRF for a range of running speeds. Since the MSD-model parameters associated with the lower mass and spring are crucial to predict GRF (Eq. 7), this may open opportunities to use segmental kinematics and/or accelerometry from lower extremities to estimate MSD-model parameters. This does however require that the lower limb accelerations measured from e.g., a tibia-mounted accelerometer are similar to the MSD-model's lower mass acceleration required to accurately predict GRF, something which is not a given. Recent studies have for example shown promising results in predicting GRF during sprinting, in high level sprinters, when contact and flight time, in combination with kinematics from the ankle were used as input for a two-mass model (*Udofa, Ryan & Weyand, 2016*; *Clark, Ryan & Weyand, 2017*). Future studies are however still need to explore the use of body-worn micro sensor technology to drive simple human body models and predict GRF waveforms for a range of running speeds.

## Model limitations

A limitation with the MSD-model and the associated model parameters is that multiple parameter combinations exist when fitting the MSD-model to measured TrunkAcc or GRF waveforms. Whilst it could be of interest to further explore the physical meaning of the individual model parameters (ACC$_{param}$ or GRF$_{param}$) and their interactions, or within and between subject parameters variations, this was not possible due to the existence of multiple model parameter solutions. Trunk-mounted accelerometry has a major benefit that it is already in use in many field contexts, but a limitation is that it may not very well represent the acceleration of the trunk segment. We have in previous work (*Nedergaard et al., 2017b*) shown that vertical trunk accelerations, measured from a high-end lab-based motion capture system, improved the upper mass acceleration fitting (median RMSE: 0.03 g across all running speeds) and lowered the average median RMSE of the GRF predictions to 5.18 N kg$^{-1}$ (vertical GRF) across all running speeds, compared to 8.99 N kg$^{-1}$ in the current study. Importantly, the accuracy and reliability of the GRF predictions are considered poor in both cases, suggesting that our hypothesis that the MSD-model's upper mass acceleration primarily represents the trunk acceleration is most likely the weakest link. Secondly, the MSD-model is a one-dimensional model, and therefore only allows the magnitude of the resultant GRF to be estimated. We decided to predict the magnitude of the resultant GRF in our study, considering that we wanted to estimate the overall external

biomechanical loading on the body, however we accept that others may prefer to predict the magnitude of the vertical GRF only. Ultimately, we believe that it is important to recognise that the MSD-model approach omits any direction specific load variations across running speeds, and that these may well be relevant in how the musculoskeletal tissues are exposed to stresses. Finally, the MSD-model is a passive elastic model and therefore does not account for additional energy generated by the body's "active" structures (muscles). Whilst a more complex model could account for this (*Zadpoor & Nikooyan, 2010*; *Nikooyan & Zadpoor, 2011*), it is questionable if this would allow for better GRF predictions from TrunkAcc. The complexity of such model would probably also defeat the overall purpose of using a simple model that is still applicable in field settings.

## CONCLUSIONS

In this study, we demonstrated that the upper mass acceleration of a simple MSD-model can be fitted to measured trunk accelerometry signals with high accuracy during running at various speeds, but that the ensuing $ACC_{param}$ do not deliver accurate predictions of GRF waveforms. Despite the convenient hypothesis that the MSD-model's upper mass acceleration primarily represents the acceleration of the trunk, our results showed that this hypothesis is violated too much to still predict meaningful GRF waveforms. Nevertheless, further studies should continue to explore the use of data from wearable micro sensor technology to drive simple human body models that could allow us to estimate GRF waveforms in field settings. This would allow researchers and practitioners to better monitor the external biomechanical loads to which the human body is exposed during running locomotion, ultimately supporting a general quest towards field-based monitoring of tissue load-adaptation processes.

## ACKNOWLEDGEMENTS

The authors would like to thank Ms Elena Eusterwiemann for her assistance with the data collection.

### Funding

This study was funded by the Football Exchange, Research Institute for Sport and Exercise Sciences, Liverpool John Moores University, UK, and partially supported by the UEFA Research Grant Programme 2014. There was no additional external funding received for this study. The funders had no role in study design, data collection and analysis, decision to publish, or preparation of the manuscript.

### Grant Disclosures

The following grant information was disclosed by the authors:
Football Exchange, Research Institute for Sport and Exercise Sciences, Liverpool John Moores University, UK.
UEFA Research Grant Programme 2014.

## Competing Interests

The authors declare there are no competing interests.

## Author Contributions

- Niels J. Nedergaard conceived and designed the experiments, performed the experiments, analyzed the data, contributed reagents/materials/analysis tools, prepared figures and/or tables, authored or reviewed drafts of the paper, approved the final draft.
- Jasper Verheul and Barry Drust conceived and designed the experiments, authored or reviewed drafts of the paper, approved the final draft.
- Terence Etchells contributed reagents/materials/analysis tools, approved the final draft, developed the mass-spring-damper model optimisation routine.
- Paulo Lisboa, Mark A. Robinson and Jos Vanrenterghem conceived and designed the experiments, contributed reagents/materials/analysis tools, authored or reviewed drafts of the paper, approved the final draft.

## Human Ethics

The following information was supplied relating to ethical approvals (i.e., approving body and any reference numbers):

The institutional ethics committee at Liverpool John Moores University granted Ethical approval (ethics approval number: 09/SPS/010) to carry out the study within its facilities.

## Data Availability

The raw date for all trials: measured and modelled TrunkAcc and GRF waveforms; estimated MSD-model parameters ($ACC_{param}$ and $GRF_{param}$) can be found online at https://doi.org/10.6084/m9.figshare.7110668.v1.

## Supplemental Information

Supplemental information for this article can be found online at http://dx.doi.org/10.7717/peerj.6105#supplemental-information.

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
