# Peer review of "The feasibility of predicting ground reaction forces during running from a trunk accelerometry driven mass-spring-damper model"

_PeerJ, doi:10.7717/peerj.6105_

## Round 0.1 · original submission · Major Revisions

Apologies for the delay, but we now have 2 peer reviews. 1 is brief with small revisions needed. 1 is much more in depth and will require moderate revisions and re-review by this reviewer. But their requests all seem manageable and the paper should improve markedly. I thank both reviewers for their timely reviews (obtaining reviewers was the cause of delay, not the reviews themselves!).

I also encourage the authors to include all data necessary to replicate/expand on this study, e.g. in a data repository.

I look forward to your revised version of this MS, which does appear to be a strong, publishable study.

Reviewer 1 ·

Basic reporting

no comment

Experimental design

no comment

Validity of the findings

no comment

Additional comments

General comments
This study investigated the accuracy of MSD model to calculate trunk acceleration and to estimate GRF with the obtained trunk acceleration. This is an interesting study, and generally well written. I enjoyed reading this manuscript. One major concern is that the acceleration measured using trunk mounted accelerometer may not represent the actual trunk acceleration. If the acceleration measured using trunk mounted accelerometer is not equal to the actual trunk acceleration measured using a motion capture system or other device, inaccurate estimation of GRF possibly came from the inaccurate trunk acceleration.

Specific comments
L98: Please add the velocity of running in this previous study.

L126: “2 m from the centre of the force platform” is not clear. Please specify the locations of the photocell gates.

L282: Please eliminate this last sentence.

Reviewer 2 ·

Basic reporting

Model naming. The authors estimate model parameters from measured trunk acceleration as well as from measured GRFs. It becomes somewhat confusing in the results and discussion to keep track of which source data were used to estimate the parameters (example line 214/215. These parameters were calculated using GRFs rather than truckacc). Suggest using an abbreviation or some sort of means to distinguish which model parameters are being discussed (GRFparameters or ACCparameters). Along these lines, including a table that lists the specific model parameters determined by each model explored would be very helpful.

Literature references well represented. One omission is in the paragraph starting on line 65. The authors state that monitoring GRFs are limited to the lab. Line 71 alludes to estimating GRFs but does not include a reference. I suggest including references here (Full disclosure that I’m suggesting referencing work I’m directly involved with (https://round-lake.dustinice.workers.dev:443/https/doi.org/10.1371/journal.pone.0099023). While this may be biased, I do believe that this reference is helpful in this article since GRFs are estimated from acceleration using statistical methods. This method is currently limited to peak vertical GRFs only. The work the authors presented here is targeting complete resultant GRF waveforms from acceleration and therefore is of interest to folks working in the area of portable methods to quantify GRFs.)

Line 78 and 85: Sentences on these 2 lines seem to contradict each other. Suggest clarifying wording. Line 78 states that a single accelerometer is inadequate to derive GRF waveforms while line 85 proposes a single sensor would be a useful tool. Possibly add to line 78 that ‘to date’ a single sensor has been inadequate but would be brilliant if feasible…

Figure 3: Suggest including how ‘extreme outliers’ were determined? How many of these outliers were excluded?

Figures 3 and 4: Include information about the source of the representative data. Single subject? Multiple subjects? Single stride? Average of multiple strides?

Experimental design

Figure 1 illustrates the overall MSD model. If the acceleration from the trunk (a1) was measured, why wasn’t the acceleration of the leg (a2) measured as well to further enhance the parameter estimations? The purpose of the approach proposed in the paper is to have a method to estimate GRF that does not need a FP. Adding a second accelerometer/GPS unit seems to maintain that purposes while also providing an ‘easy’ means to quantify additional variables from the model.

Line 121: Add a space after ‘approval’ and before ‘for’.

Line 121: No details were included regarding the warm up period. Please provide information about the warm up completed by subjects.

Line 127: The use of 4 trials at each speed seems small. Suggest adding information about how 4 trials was the chosen number of trials to complete. Additionally, how were unsuccessful contacts with the force plate handled? Were these data included? Trial repeated?

Line 128: Please include how dominant leg was determined.

Line 135: Suggest including more information about the Minimax device used. How does it report raw accelerations? Are they processed using proprietary algorithms?

Line 137: For this study, the GPS device was placed on the dorsal part of the upper trunk. Suggest adding information about how this location was chosen. Seems like this position may be subjected to more rotational accelerations that other locations on the trunk? What steps were done to ensure tight coupling between the elastic vest and the subject

Line 142: Assuming the last sentence is referring to only stance phase data being collected for GRFs? If so, please specify in the sentence.

Line 163 and on: Please specify that resultant GRF was estimated. (for example, line 170 states ‘..GRF acting on the MSD-model…;. This should specify resultant GRF.)

Line 176: You specify that ‘for each trial’ model parameters were optimized. In subsequent sentences and figures, it’s unclear if the calculations being discussed are also for each trial or were potentially averaged/grouped somehow (example line 183 discussing GRF predictions). Please clarify possibly with a general sentence early in the methods that all calculations discussed were for each trial. Similarly, the figures include ‘representative’ examples but it is unclear if these examples are for a particular subject, several subjects, all subjects? Please include more details.
Follow up question: If I understood correctly, each trial was analyzed independently. Was there any analysis done to look at variations in the parameters calculated for a subject across his 4 trials? Between subjects differences?

Validity of the findings

Overall a well thought out and thorough discussion of the results. One point that keeps arising for me is it seems that based on your results, ‘like predicts like’. Model parameters predicted from GRFs, predict GRFs well. Model parameters predicted from accelerations, predict accelerations well. The predicted GRFs in Figure 3 look like very similar to the accelerations with a phase lag, yes? After reviewing Figure 3 and reading through the explanations as to why the model predictions failed when using parameters identified from the trunkacc to predict GRFs, I’m left to wonder if it’s just a simple ‘like predicts like’ versus the trunk accelerations measured do not represent the a1? Along these lines, it seems it would be useful to explore the overall feasibility and precedent (if there is any?) of predicting GRFs with accelerations or predicting accelerations with GRFs?


Line 332: change ‘is’ to ‘are’

Additional comments

Generally, the article is very well written and fairly clear, especially considering the number of variables and complexity of the model(s) explored.

---

## Round 0.2 · Minor Revisions

Apologies for delay in getting this final review, but it was crucial and the main point made about supplying parameter descriptions is important. Please make final amendments-- there should be no need for further review; I will check that changes are satisfactory. Thank you!

Reviewer 2 ·

Basic reporting

The authors have addressed previous comments and questions very well. I appreciate the time and effort put into editing the manuscript to improve the clarity. One question/comment remains for me.

The 8 model parameters. Thank you for including abbreviations to distinguish the source of the model parameters. This helped clarify immensely for me. One issue remains however. As currently written, there is no way for the reader to have any idea of the differences in the model parameters, even a ballpark difference. Since the specific parameters are critical to the estimations, it seems as though these should be reported. While I understand the authors concerns about potentially confusing readers by reporting two sets of these parameters, for transparency to the reader that might be interested in these values, the parameters should be reported. If another researcher is attempting to apply these findings to their own outcomes, the parameter specifics are critical.

Experimental design

No comments.

Validity of the findings

Please include some details of how RMSE was calculated. Was every data point used to calculate overall RMSE? Peak values?

Additional comments

No comments.

---

## Round 0.3 · accepted · Accept

Excellent- this is what was needed, I think. The paper clearly is ready. Thank you again for your patience!

#